The mineralization characteristics of organic carbon and particle composition analysis in reconstructed soil with different proportions of soft rock and sand

Guo Zhen 1 2 3 4
Han Jichang 1 2 3 4
Xu Yan 1 2
Lu Yangjie 1 2
Shi Chendi 1 2
Ge Lei 1 2
Cao Tingting djyjygz2019@139.com 1 2 3 4
Li Juan 2644816206@qq.com lijuan8136@163.com 1 2 3 4
1 Shaanxi Provincial Land Engineering Construction Group Co., Ltd./Shaanxi Key Laboratory of Land Consolidation , Xi’an , Shaanxi , China
2 Institute of Land Engineering and Technology, Shaanxi Provincial Land Engineering Construction Group Co., Ltd. , Xi’an , Shaanxi , China
3 Key Laboratory of Degraded and Unused Land Consolidation Engineering, the Ministry of Natural and Resources of China , Xi’an , Shaanxi , China
4 Shaanxi Provincial Land Consolidation Engineering Technology Research Center , Xi’an , Shaanxi , China
Kuwae Tomohiro
Electronic publication date: 2019 Sep 16
Publication date: 2019
Volume: 7
Electronic Location ID: e7707
Received 2019 May 29; Accepted 2019 Aug 20
Copyright: ©2019 Guo et al.
Copyright year: 2019
Copyright holder: Guo et al.
License: This is an open access article distributed under the terms of the Creative Commons Attribution License, which permits unrestricted use, distribution, reproduction and adaptation in any medium and for any purpose provided that it is properly attributed. For attribution, the original author(s), title, publication source (PeerJ) and either DOI or URL of the article must be cited.
License URL: https://creativecommons.org/licenses/by/4.0/

Keywords: Microstructure, Texture, Organic carbon mineralization, Soft rock, Eolian sand soil

Funding: The Fundamental Research Funds for the Central University, CHD 300102279503 TFund Project of Shaanxi Key Laboratory of Land Consolidation 2018-JC18 2019-JC07 Shaanxi Provincial Land Engineering Construction Group internal research project DJNY2019-12 DJNY2018-12 This study is financially supported by the Fundamental Research Funds for the Central University, CHD (300102279503), the Fund Project of Shaanxi Key Laboratory of Land Consolidation (2018-JC18, 2019-JC07) and the Shaanxi Provincial Land Engineering Construction Group internal research project (DJNY2019-12, DJNY2018-12). The funders had no role in study design, data collection and analysis, decision to publish, or preparation of the manuscript.

==============================
The organic carbon mineralization process reflects the release intensity of soil CO2. Therefore, the study of organic carbon mineralization and particle composition analysis of soft rock and sand compound soil can provide technical support and a theoretical basis for soil organic reconstruction (soil structure, materials and biological nutrition). Based on previous research, four treatments were selected: CK (soft rock:sand=0:1), C1 (soft rock:sand=1:5), C2 (soft rock:sand=1:2) and C3 (soft rock:sand=1:1), respectively. Specifically, we analyzed the organic carbon mineralization process and soil particle composition by lye absorption, laser granulometer, and scanning electron microscope. The results showed that there was no significant difference in organic carbon content between C1, C2 , and C3 treatments, but they were significantly higher than in the CK treatment (P < 0.05). The organic carbon mineralization rate in each treatment accords with a logarithmic function throughout the incubation period (P < 0.01), which can be divided into a rapid decline phase in days 1 to 11 followed by a steady decline phase in days 11 to 30. The cumulative mineralization on the 11th day reached 54.96%–74.44% of the total mineralization amount. At the end of the incubation, the cumulative mineralization and potential mineralizable organic carbon content of the C1, C2 and C3 treatments were significantly higher than those of the CK treatment. The cumulative mineralization rate was also the lowest in the C1 and C2 treatment. The turnover rate constant of soil organic carbon in each treatment was significantly lower than that of the CK treatment, and the residence time increased. With the increase of volume fraction of soft rock, the content of silt and clay particles increased gradually, the texture of soil changed from sandy soil to sandy loam, loam , and silty loam, respectively. With the increase of small particles, the structure of soil appear ed to collapse when the volume ratio of soft rock was 50%. A comprehensive mineralization index and scanning electron microscopy analysis, when the ratio of soft rock to sand volume was 1:5–1:2, this can effectively increase the accumulation of soil organic carbon. Then, the distribution of soil particles was more uniform, the soil structure was stable (not collapsed), and the mineralization level of unit organic carbon was lower. Our research results have practical significance for the large area popularization of soft rock and sand compound technology.

Introduction

The farmland soil organic carbon pool plays an important role not only in the process of global carbon circulation but also as the most important material base for soil fertility (Pan, Smith & Pan, 2009), and has a decisive role among others in the maintenance of cultivated land productivity, the prevention and treatment of soil erosion, the spatial and temporal variation of soil respiration, and its stability (Tommaso et al., 2018). The soil organic carbon pool in terrestrial ecosystems is about three times that of the plant carbon pools (Lal, 2004). The organic carbon exchanged between soil and atmosphere accounts for about 2/3 of the total carbon storage of surface ecosystems, which slightly changed in recent years because of the impact of greenhouse gas emissions (Lal, 2004). Soil organic carbon mineralization is an important part of the carbon cycle in terrestrial ecosystems. The change in land resources by human activities causes changes in atmospheric CO2 concentration through the effects on terrestrial ecosystems which in turn affects the carbon cycle and the climate change process (Stumpf et al., 2018). Therefore, the research on soil carbon cycle by human activities has attracted much attention in recent years and has become the core issue of multidisciplinary research (Feng et al., 2006; Schmitz et al., 2017; Zhang et al., 2018).

Soil organic carbon mineralization is a process in which organic substances are decomposed into inorganic substances by microbial degradation, providing nutrients for crop growth, and releasing greenhouse gases such as CO2 (Dai, Wang & Fu, 2017; Guo et al., 2018). Current studies have shown that arid and semi-arid regions account for 41% of the global land area, carrying 38% of the human population, and which are sensitive to global climate change and human activities (Zhou & Zhang, 2009). Experts call it the arid and semi-arid areas of Shaanxi, Shanxi, Inner Mongolia and other arid areas “Earth Environment Cancer”, since these important soil resources exhibit serious soil and water loss, loss of soil texture, low nutrient content and poor structure. This can be seen as a geographical hazard of the soft rock and sand areas and it determines the urgency and difficulty of ecological restoration in the region (Wang et al., 2007). Han, Liu & Luo (2012) studied the structure and physicochemical properties (including capillary porosity, saturated hydraulic conductivity, organic matter, water-stable aggregates, mineral composition, and crop yield) of soft rock and sand and realized that the two soil forms can be mixed into different proportions to form a “new soil”. They also suggested that the optimal compounding ratio for crop growth ranged between 1:5 and 1:1 (soft rock: sand). Presently, the technology of compounding of soft rock and sand into soil is widely used, and the newly added cultivated land amounts to 1,573 ha, which realized the resource utilization of soft rock and improved the regional ecological environment (Han, Liu & Luo, 2012). She et al. (2015) studied the nutrient content and hydraulic parameters of the soft rock and sand compound soil, and proposed that the fertilizer retention performance of the compound soil can be effectively improved with an increase of the soft rock content, whereas the available nutrient content can be improved with sufficient water. Wang (2016) mixed sand and soft rock to a certain proportion forming a new type of soil, analyzed its textural changes and biochemical indexes of crops, and discussed the soil texture changes from sand to silt loam as the proportion of soft rock increased. Moreover, the photosynthesis rate of crops and the activity indexes of antioxidant enzymes increased at the beginning and then decreased. However, considering the present global warming, we should not only improve the utilization rate of waste resources but also maintain the sustainable development of ecological environment. The issue of greenhouse gas emissions from the compounded soil is an area which has not yet been considered by researchers. The carbon pool balance depends not only on carbon emissions but also on carbon sequestration. Therefore, the carbon source or sink of the composite soil also needs to be further studied.

The texture of soft rock is loose because of the complementary nature of its components: the large amount of montmorillonite is strongly hydrophilic with a high adsorption potential whereas the aeolian sandy soil is leaking water and fertilizer (Sun & Han, 2018). Research on the soft rock and sand compound soil in the early stage showed that the increase of the ratio of soft rock to sand can effectively increase the capillary porosity and decrease the water infiltration coefficient of the aeolian sandy soil (Li, Li & Wang, 2018; Shen et al., 2013; Wang et al., 2017; Zhang et al., 2016). The texture was also improved to a certain extent from sand to sandy loam (Wang et al., 2017). The results of the mechanical test (conditions: the loading confining pressure was set to 50, 100, 200 kPa; the moisture content of the compound soil was 14%; the dry density was 1.6 g cm−3) for different mixing ratios of soft rock and sand using a dynamic three axis showed that the macroscopic mechanics exhibited a strain-hardening phenomenon with non-linear characteristics (Lei, Wang & Xie, 2018). After adding soft rock, the final water content of the improved soil was significantly higher than that of the aeolian sandy soil, which was beneficial for the maintenance of water and fertilizers. The results also indicated that the toxic effect of lead in aeolian sandy soil is effectively and proportionally reduced by the amount of soft rock added (Li, Li & Wang, 2018). However, the research only focused on the hydraulic properties, fertility, and the adsorption of the soft rock and aeolian sandy soil during the early stage, lacking research on the carbon source/carbon sink effect of the compound soil. Therefore, the purpose of this study is to: (1) clarify the carbon mineralization strength of the compound soil in different proportions of sandstone and sand; (2) understand the microstructure and particle composition of the compound soil; and (3) clarify the carbon fixation effect of different proportions of mixed soil and provide a basis for a sustainable development of the regional ecology. We hypothesized that (1) the addition of different proportions of soft rock can effectively increase the soil organic carbon and mineralized carbon content, acting as a carbon sink when the volume fraction of soft rock is less than 50%; and (2) with the increase of volume fraction of soft rock, the soil fine particles increase, changing the texture from sand to silt loam.

Materials & Methods

Overview of the test site

The experimental plot was set up at the Fuping County pilot test base from the Shaanxi provincial land engineering technology research institute. Fuping County (108°57′−109°26′E, 34°42′−35°06′N) lies at the transition zone between the Guanzhong plain and the northern Shaanxi plateau that belongs to the gully region of the Weibei Loess Plateau. The terrain is high in the north and low in the south. It slopes from the northwest to the southeast. The elevation in the territory is between 376–1,421 m. The area belongs to the continental monsoon warm zone with a semi-arid climate. The annual total radiation is 5,187 MJ m−2, the annual average sunshine hours are about 2,389 h with an annual average temperature of 13 °C, and an annual average precipitation of 527 mm. The interannual variation of precipitation is high, and the annual precipitation coefficient of variation (CV) reaches 21%.

Experiment design

The field test plot was set to simulate the land condition of the soft rock and sand mixed layer in the Mu Us sandy land. The experimental plot layers were set to a mixture of soft rock and sand at 0–30 cm, and a layer of aeolian sandy soil between 30–70 cm. The soft rock and sand were taken from the Daji Han Village, Xiaoji Han township, Yuyang district, and Yulin city. The minerals in the soft rock mainly include quartz, montmorillonite, feldspar, calcite, illite, kaolinite, and dolomite. The main chemical constituents of soft rock are SiO2 (65% by mass), Al2O3 (14% by mass), Fe2O3 (12% by mass) and CaO (9% by mass). The mineral in the sand is mainly quartz (SiO2) with a mass fraction of about 82%. The remaining minerals are mainly feldspar (10% by mass), kaolinite (4% by mass), calcite (2% by mass), and amphibole (2% by mass). The organic carbon content of soft rock and sand are 2.00 g kg−1 and 0.63 g kg−1, respectively.

The analysis was performed in 2009, and four treatments of soft rock and sand with the different volume ratios of 0:1 (CK), 1:5 (C1), 1:2 (C2), and 1:1 (C3) were selected. Each treatment was repeated 3 times with a total of 12 trial plots and randomly distributed. Each plot area covered 2 m × 2 m (= 4 m2). For uniformity of factors such as illumination and micro-topography, the test plot was arranged from south to north in a “one” shape with a depth of the soil layer of 30–40 cm (Fig. 1), whereas the mixing depth of the soft rock and sand in the test plot was designed in the first 30 cm, simulating the field conditions. The layer between 30–70 cm was completely filled with sand. The experimental field crops were corn (Jincheng 508) and wheat (Xiaoyan 22), thus providing two different crops a year, all of which were artificially sown. Different types of fertilizers were tested in the experimental field consisting of urea (including N 46.4%), diammonium phosphate (containing N 16% and P2O5 44%), and potassium sulfate (including K2O 52%). The amount of fertilizer applied was 255 kg ha−1(N), 180 kg ha−1(P2O5) and 90 kg ha−1(K2O). All treatments were similarly applied. All phosphate and potassium fertilizers were used as base fertilizers, 65% of nitrogen fertilizers were used as base fertilizers, and 35% were combined with the irrigation water to be applied at the booting stage. Wheat planting time is generally in the middle and late October, whereas the corn planting time is in the middle and late May. One to two days before sowing, the three fertilizers were weighed according to the required amount of each plot, evenly spread on the soil surface, and then properly ploughed to mix the fertilizer with the topsoil.

Figure 1 Test plot layout for soft rock and sand compound soils.

Soil sample collection

After the wheat harvest in May 2018, five uniformly distributed samples from the 0–30 cm soil layer of each plot were collected to form a mixed sample. Animals and plant residues were removed from the collected soil samples. Afterwards, the samples were sieved through a 2 mm sieve and divided into two parts, one part was placed in a 4 °C refrigerator for the mineralization incubation test whereas the other part was naturally air-dried, ground and then sieved through a 1 mm and a 0.149 mm screen for a subsequent scanning electron microscopy analysis to determine the texture and organic carbon.

Determination method

The mineralization incubation test was carried out by the Alkali Absorption Method (Dai, Wang & Fu, 2017; Ribeiro et al., 2010; Zibilske, 1994) ). For this, the soil placed in the refrigerator at 4 °C was weighed and 30 g transferred into a beaker, set to a field water holding capacity of about 60%, and then pre-incubated in the incubator at 25 °C for 5 days. The purpose was to restore microbial activity and adapt to the current incubation environment. Then, the lye and the beaker filled with the soil were transferred into an incubation bottle, sealed, and incubated in the dark. The lye was then analyzed by titration with diluted hydrochloric acid at the days 1, 2, 3d, 4, 5, 8, 11, 14, 18, 22, 26, and 30, respectively (borax (0.05 mol L−1) was used for calibration before each titration). At each titration, 2 mL of 1 mol L−1 BaCl2 solution was added to the soil-filled beaker together with 2 drops of phenolphthalein indicator. Water was added to the soil to a constant weight each time the lye absorption cup was changed.

Organic carbon was determined by the potassium dichromate-concentrated Sulfuric Acid External Heating method (Nelson & Sommers, 1996). The texture was measured using a Malvern laser particle size analyzer (MS2000; Malvern Instruments, Malvern, UK). The soil sample after the 1 mm sieving was cured by epoxy resin, first coarsely then artificially ground, and polished by a sander to make the surface smooth. Thus prepared, a microsample with a diameter of 5 mm and a height of 3 mm was obtained. The dried sample was subjected to gold plating, and scanning was performed by a scanning electron microscope (FEI Q45, USA) using a voltage of 25 kV in an “S” type and a magnification of 1,000 times.

Data processing and analysis

The cumulative amount of soil organic carbon mineralization refers to the total amount of soil CO2 released from the beginning of cultivation to a certain time point. It can be fitted by the first-order kinetic equation using the Origin drawing software 2017, i.e., Ct = C0 (1-e−kt) (Ribeiro et al., 2010), (1) Ct=C01−e−kt

Where Ct is the accumulated mineralization amount of soil organic carbon in mg kg−1after time t; C0 is the soil potential mineralized organic carbon in mg kg−1; k is the constant of organic carbon mineralization rate, d−1; t is the number of days of cultivation, and d the half-turn period T1∕2 = ln2/k.

Texture data was classified using the TriangleVB software. All data was sorted and graphed using EXCEL 2019, and the analysis of variance and multiple comparisons were performed using SPSS 19.0.

Results

Compound soil organic carbon

The organic carbon content of aeolian sandy soil can be significantly improved by adding different proportions of soft rock (Fig. 2). The organic carbon content in the CK treatment was 2.02 g kg−1. The organic carbon content in the C1, C2, and C3 treatments was significantly increased compared to that of the CK treatment (P < 0.05) with increases of 110%, 77% and 119%, respectively. With the increase of the proportion of soft rock, the soil organic carbon first increased, then decreased, to continue to increase again. The organic carbon content in the C1 treatment was 4.24 g kg−1, and the organic carbon content of C2 and C3 treatment decreased and increased by 16% and 4%, respectively. Though there was an increase by 24% in the C3 treatment compared to the C2 treatment, there was no significant difference (P > 0.05).

Figure 2 Organic carbon content of compound soils in different proportions of soft rock and sand.

Different letters above the bars mean significant difference (at 0.05 level) between treatments. CK, the volume ratio of soft rock to sand is 0:1; C1, the volume ratio of soft rock to sand is 1:5; C2, the volume ratio of soft rock to sand is 1:2; C3, the volume ratio of soft rock to sand is 1:1.

Compound soil organic carbon mineralization rate

The mineralization rate of organic carbon in soils with different mixing ratios of soft rock and sand showed a logarithmic, dynamic downward trend with the cultivation time with a relation of y = a + b ln(x) (P < 0.01) (Fig. 3, Table 1). The logarithm function does not introduction of mineralization potential and other parameters, which reflects the pure digital variation law, indicating that the data trend is better. The mineralization rates of organic carbon in the CK and C3 treatments reached their peaks compared to day 1 on day 3 of incubation with 49.6 mg kg−1 d−1 (76%) and 66.1 mg kg−1 d−1 (37%), respectively. The mineralization rate of organic carbon decreased rapidly after day 3 slowing down its decline after day 11. The organic carbon mineralization rate at 30 days of cultivation was 3.4 mg kg−1 d−1 (CK) and 13.5 mg kg−1 d−1 (C3), respectively, which indicates a decrease by 93% and 80% compared to day 3. However, the organic carbon mineralization rate in the C1 and C2 treatment exhibited its maximum on the first day of incubation with 58.7 mg kg−1 d−1 and 43.5 mg kg−1 d−1, respectively. The mineralization rate in the C1 and C2 treatments rapidly declined until day 11, after that this decline slowed down. The mineralization rate on the day 30 was 83% and 81.68% lower than that of the first day, respectively. Taken together, the average mineralization rate in the C3 treatment was the highest in all compound ratio treatments, followed by the C1 and the C2 treatment, whereas the CK treatment exhibited the lowest mineralization rate. The mineralization rate of all treatments can be divided into two stages: namely a rapid decline (1–11 days) and steady decline (11–30 days). In general, the CO2 production rate changed greatly during the rapid decline phase, whereas the mineralization rate between all treatments was consistent during the steady decline phase.

Figure 3 Organic carbon mineralization rate of compound soils in different proportions of soft rock and sand.

CK, the volume ratio of soft rock to sand is 0:1; C1, the volume ratio of soft rock to sand is 1:5; C2, the volume ratio of soft rock to sand is 1:2; C3, the volume ratio of soft rock to sand is 1:1.

Table 1 Regression equation of soil organic carbon mineralization rate and cumulative mineralization under different compounding ratios.

Index	Treatment	Regression equation	r	
Mineralization rate	CK	y1= − 12.10 ln(x) + 41.28	0.7933**	
C1	y1= − 14.56 ln(x) + 54.378	0.9298**	
C2	y1= − 10.06 ln(x) + 40.408	0.9595**	
C3	y1= − 14.40 ln(x) + 64.713	0.8679**	
Cumulative mineralization	CK	y2=71.928 ln(x) + 30.107	0.9935**	
C1	y2=132.40 ln(x) + 9.2284	0.9752**	
C2	y2=121.50 ln(x) − 14.676	0.9757**	
C3	y2=234.98 ln(x) − 60.922	0.9795**	
** means significant correlation at 0.01 level.

CK the volume ratio of soft rock to sand is 0:1

C1 the volume ratio of soft rock to sand is 1:5

C2 the volume ratio of soft rock to sand is 1:2

C3 the volume ratio of soft rock to sand is 1:1

y1 CO2 production rate, mg kg−1d−1

y2 CO2 accumulative release amount, mg kg−1

x incubation day, d

Cumulative mineralization of compound soil organic carbon

The relationship between the cumulative mineralization of organic carbon and the incubation time in different proportions of soft rock and sand demonstrated a logarithmic function relationship y = a + b ln(x) (P < 0.01) (Fig. 4, Table 1). The results showed that the organic carbon accumulation mineralization decreased gradually with the incubation time, which indicates that the CO2 release rate decreased. During the whole incubation period, the cumulative mineralization of organic carbon was significantly the highest in the C3 treatment, followed by the C1 and C2 treatments, whereas the CK treatment had the lowest mineralization accumulation (F = 26.54, P < 0.01). After incubation for 30 days, the cumulative mineralization of organic carbon in the CK treatment was 274 mg kg−1. The cumulative mineralization of organic carbon treated by C1, C2 and C3 increased significantly by 88%, 59%, and 193% as compared to CK, respectively (Table 2). There was no significant difference in the cumulative mineralization of organic carbon between the C1 and C2 treatments. Compared to the C1 and C2 treatment, the cumulative mineralization of organic carbon in C3 treatment was significantly increased by 55% and 84%, respectively.

Figure 4 Organic carbon cumulative mineralization of compound soils in different proportions of soft rock and sand.

CK, the volume ratio of soft rock to sand is 0:1; C1, the volume ratio of soft rock to sand is 1:5; C2, the volume ratio of soft rock to sand is 1:2; C3, the volume ratio of soft rock to sand is 1:1.

Table 2 Cumulative mineralization of SOC after the 30 days of incubation and parameters of its kinetic equations.

Treatment	Ct (mg kg−1)	C0 (mg kg−1)	k (d−1)	T1/2	C0/SOC (%)	R2	
CK	274.44 c	257.44 c	0.1671 a	4.15 b	12.78 b	0.9748**	
C1	517.03 b	526.05 b	0.0824 b	8.41 a	12.42 b	0.9643**	
C2	437.22 b	484.61 b	0.0697 b	9.94 a	13.61 b	0.9947**	
C3	803.88 a	936.95 a	0.0622 b	11.14 a	21.25 a	0.9983**	
** indicates a extremely significant level of 1%.

CK the volume ratio of soft rock to sand is 0:1

C1 the volume ratio of soft rock to sand is 1:5

C2 the volume ratio of soft rock to sand is 1:2

C3 the volume ratio of soft rock to sand is 1:1

Ct for amount of organic carbon cumulative mineralization, C0 for amount of potential mineralizable organic carbon, k for constant of organic carbon mineralization rate, T1∕2 for half turnover period, C0/SOC for ratio of potential mineralizable organic carbon to total organic carbon in compound soil. Values followed by different letters in the same column mean significant difference at 0.05 level between treatments.

Cumulative mineralization rate of organic carbon in compound soil

The cumulative mineralization rate of soil organic carbon in the different compound ratios of soft rock and sand can reflect the strength of the carbon fixation capacity in the new compound soil. The higher the ratio, the weaker the carbon sequestration capacity of the soil, and vice versa. Figure 5 shows that the cumulative mineralization rate of soil organic carbon in the three treatments of CK, C1, and C2 did not reach a significant difference after 30 days of incubation (P > 0.05). In our analysis, C1 exhibited the lowest cumulative mineralization rate in the three treatments. The cumulative mineralization rate of organic carbon in the C3 treatment was 18%, which was significantly increased by 4.6, 6.0 and 6.0 percentage points as compared to CK, C1 and C2, respectively. Compared to the C1 treatment, the cumulative mineralization rate of organic carbon in CK, C2, and C3 treatments increased by 1.4, 0.1 and 6.0 percentage points, respectively.

Figure 5 Cumulative mineralization rate of soil organic carbon under different compound ratios during the 30 days of incubation.

CK, the volume ratio of soft rock to sand is 0:1; C1, the volume ratio of soft rock to sand is 1:5; C2, the volume ratio of soft rock to sand is 1:2; C3, the volume ratio of soft rock to sand is 1:1. Different letters above the bars mean significant difference (at 0.05 level) between treatments.

Fitting parameters of organic carbon mineralization in compound soil

There were significant differences between the parameters of the kinetic equations of organic carbon mineralization in the treatments with different proportions of soft rock and sand. Indeed, the first-order kinetic equation Ct = C0 (1-e−kt) was used for parameter fitting (P < 0.01) and introduced two parameters of mineralization rate constant and potential mineralizable carbon. The potential mineralizable organic carbon (C0) content of the CK treatment was 257 mg kg−1, and the C0 values in C1, C2 and C3 treatment were significantly increased by 104%, 88%, and 264%, respectively (P < 0.05). There was no significant difference between the C1 and C2 treatments (Table 2), whereas the C3 treatment exhibited significantly increased C0 values with 78% and 93% compared to the C1 and C2 treatments. The k-values (the organic carbon mineralization rate constant) of the C1, C2, and C3 treatments were lower than those of the CK treatment though not significantly. The trend of T1∕2(half-turn period) was opposite to that of the k-values, indicating that the addition of different proportions of soft rock reduced the mineralization rate constant of organic carbon and increased the retention time of organic carbon in the soil. The C0/SOC (SOC: soil organic carbon) of the CK treatment was 13%. Taken together, the C1, C2, and CK treatments were not significantly different to each other whereas the C3 treatment was significantly increased by 8% compared to the CK treatment.

Compound soil microstructure

The microstructure of the compound soil provides us some information about the corresponding soil structure. Using the scanning electron microscope (SEM) we observed irregular shapes of sand grains. Though the degree of grinding was high, there were no sharp edges and angles in the CK treatment (Fig. 6A). With the increase of the volume fraction of the soft rock (C1, C2, C3) (Figs. 6B, 6C, and 6D), the overall structure of the composite soil showed no obvious change but the filling of fine particles increased gradually. As the volume fraction of small particles gradually increased, the distance between the small and large particles increased more than the one between the large particles. When the content of soft rock reached 50% (C3, Fig. 6D), due to the increase of the specific surface area of the small particles, the large particles were not enough to support the soil structure, which entailed a collapse of the soil mass.

Figure 6 The microstructure of compound soil in different proportions of soft rock and sand.

(A) The volume ratio of soft rock to sand is 0:1 (CK); (B) the volume ratio of soft rock to sand is 1:5 (C1); (C) the volume ratio of soft rock to sand is 1:2 (C2); (D) the volume ratio of soft rock to sand is 1:1 (C3). The magnification is 1,000 times.

Compound soil mechanical composition

In the CK treatment, the content of sand was 87%. With the increase of the proportion of soft rock, the sand content gradually decreased, which was for the C1, C2, and C3 treatment as compared to the CK treatment 33, 42 and 46 percentage points, respectively (P < 0.05). The silt content increased gradually with the increase of soft rock, which was for the C1, C2, and C3 treatment an increase of 29, 35, and 39 percentage points, respectively (P < 0.05). The clay content of the CK treatment was 2.3%. With the increase of volume fraction of soft rock, the clay content in the C1, C2 and C3 treatments increased by 4.5, 7.0 and 7.2 percentage points, respectively, as compared to the CK treatment (P < 0.05). Thus, the increase of the clay component was smaller than that of silt. There were no significant differences between the three C1, C2, and C3 treatments. Taken together, with the increased volume fraction of soft rock, the texture of compound soil gradually changed from sandy soil to sandy loam, loam, and silty loam (Fig. 7).

Figure 7 The soil particle composition under different compound ratios of soft rock and sand.

CK, the volume ratio of soft rock to sand is 0:1; C1, the volume ratio of soft rock to sand is 1:5; C2, the volume ratio of soft rock to sand is 1:2; C3, the volume ratio of soft rock to sand is 1:1. Lowercase letters indicate significant differences (at 0.05 level) in the same particle composition between treatments.

Discussion

Soil organic carbon mineralization is an important biochemical process in soil, which is directly related to the release and supply of soil nutrient elements, CO2 gas emissions, and soil quality maintenance (Ross et al., 1999). In our analysis, we used four treatments, with the different volume of soft rock to sand ratios of 0:1 (CK), 1:5 (C1), 1:2 (C2), and 1:1 (C3), respectively. During the whole incubation period, the CK and C3 treatment demonstrated the same mineralization reaction characteristics, reaching a peak on the 3rd day, followed first by a rapid and subsequently a slow decline (Fig. 3). This can be explained in that the soil microenvironment was still at the beginning of the reaction and that the compound soil organic carbon in the initial stage of mineralization was mostly in the form of complex compounds. Thus, during this stage only a few small molecular compounds were easily decomposed. Microorganism need to simplify the complex compound before it can be absorbed and utilized, which is indicated by a rapid rising phase of the respiratory rate at the initial stage (Li, 2000; Alvarez et al., 1998). The reaction characteristics of the C1 and C2 treatment were similar as the trend of decrease was observed throughout the complete incubation period. All the treatments can be divided into two stages: a rapid decline on days 1 to 11 and a steady decline on days 11 to 30. The cumulative mineralization on day 11 reached 55%–74% of the total mineralization (Fig. 4), which was consistent with the study of Zhang et al. (2011). During the early mineralization stage, the organic matter mainly decomposed by soil microbes is derived from animal and plant residues and their secretions (Li, Qiu & Zhang, 2010). Thus, many active organic substances such as sugars and proteins were easily decomposed in the soil at this initial stage providing abundant carbon sources and nutrients for soil microorganisms and promoting microbial activity. With the prolongation of cultivation time, the active organic components, which were easily to decompose within the soil, were gradually used up by the microorganisms and the remaining components such as lignin and cellulose are more difficult to be access by the microorganisms (Kögel-Knabner et al., 2010; Yang et al., 2007). Therefore, the mineralization rate showed a trend from fast to slow which mirrors the decomposition rate, whereas the cumulative mineralization showed a cumulative trend of a gradual decrease in release intensity (Franzluebbers et al., 2001; Li et al., 2018). Obviously, the amount of soil nutrient plays an important role in the organic carbon mineralization process. Kemmitt et al. (2008) studied the mineralization process of some microorganisms and found that after fumigation with chloroform, which reduced the number of microorganisms by 90%, the mineralization rate of organic carbon among all treatments had no significant difference compared to the control treatment. Thus, the mineralization of soil organic matter was limited by a non-biological process that converts the substrate into microbial utilization (Kemmitt et al., 2008). At this stage, microorganisms only play a minor role as the available organic materials become a limiting factor. Based on a hypothesis, Kemmitt et al. (2008) divided the process of humified soil organic matter mineralization into two steps. The first step is a-biological and independent of any microbial processes. The possible mechanisms include chemical oxidation or hydrolysis, diffusion from inaccessible soil pores or aggregates, desorption from the solid phase, and the action of extracellular stabilized enzymes. The second step is the mineralization by the soil microorganisms of this small, now biologically available, trickle of substrate derived from humified soil organic matter. This trickle of substrate is equally available to both the small developing recolonizing population in the fumigated soil and the larger population in the non-fumigated soil. Hypothetically, if the soil microorganisms acted directly on solid soil, this would likely cause a localized depletion over long periods as most soil microorganisms are immobile. Coming back to the two-step hypothesis, the results of the first step are to convert the compounds into organic matter for microbial use, and the second step is to transfer these substances via diffusion to other microbes. Kemmitt et al. (2008) described the non-biological dominated mineralization process as similar to that of microorganisms acting directly on solid soil. Sollins, Homann & Caldwell (1996) also believe that the mineralization process and unstable processes of soil organic matter cannot be enhanced by increasing the activity of microorganisms. The mineralization process in their study may be a comparable process as that described by Kemmitt et al. (2008) for the end of the first stage and the beginning of the second stage. Taken together, a special research is needed about the role of microorganisms during the mineralization process.

Despite the same, comparable incubation conditions within our experiments, there were significant differences in soil organic carbon accumulation mineralization (Ct) with different compounding ratios, which was C3>C1>C2>CK. This indicates a consistent trend with the content of organic carbon (Fig. 2). The low content of soil organic carbon in the CK treatment affects the mineralization rate of soil organic carbon, which results in a relatively small cumulative release of CO2. The aeolian sandy soil has a higher sand content, larger permeability coefficient, and a serious water and fertilizer leakage. On the other hand, there are many small particles in the soft rock, which are hydrophilic and adsorbent. In our experiment, mixing soft rock and aeolian sandy soil in a certain proportion promoted the increase of organic carbon content and mineralized amount. The results of this study indicated that the soil clay and silt content increased with adding more soft rock. When the content of soft rock was 17% (C1 treatment), the soil texture was then a sandy loam (Fig. 7), and the cumulative mineralization rate of soil organic carbon was the lowest at the beginning (Fig. 5). When the content of soft rock was 33% (C2 treatment), the soil texture was loam, and the soil organic carbon accumulation mineralization rate was similar to the C1 treatment, indicating that a compound ratio of soft rock and sand between 1:5 to 1: 2 can promote the accumulation of soil organic carbon.

From the scanning electron microscope image (Fig. 6A) it can be seen that the soil particles in the CK treatment (aeolian sandy soil) do not adhere to each other, the spacing between the soil particles was large, and the pores developed, mainly non-capillary pores. Therefore, the aeolian sandy soil has a good ventilation effect but it has difficulties in effectively maintaining the moisture, so it will leak water and fertilizer, which was not conducive for plant growth. The soil composition of CK treatment was single and the texture uniform, most of which were sand grains. After the addition of soft rock in different proportions, the soil texture changed from sand to sandy loam to loam to silt loam (Fig. 7). Figure 6B shows that some of the single particles have a higher degree of roundness, and some soil particles with rough surface and mutual adhesion have agglomerate characteristics, which mainly derive from the soft rock. Compound soil began to appear on the surface of the clay, which was conducive to the benign transformation of the compound soil structure (Figs. 6B, 6C and 6D). The reasons for the formation of the structure can be summarized in three types: (1) When the soft rock was mixed with sand, the silt and clay particles in the soft rock came in contact with the aeolian sand, whereas the silt and clay particles were gradually adsorbed around the sand; (2) In the process of artificial improvement, irrigation and organic matter increased the cementing material in the compound soil, and the cementation promoted the formation of soil aggregates; (3) Various external stresses such as crop root activity and animal activities promote the formation of soil aggregate structure. It can be concluded that the addition of soft rock was improving the loose structure of the sandy land, promoting the agglomeration and cementation of soil, and thus improving the aeration and permeability of soil. However, adding more soft rock does not consequently lead to better results. When the soft rock content reached 50% (C3 treatment), the large particles were not enough to support the soil structure, entailing the collapse of the soil (Fig. 6D). At this composition, the soil texture was a silt loam, and the cumulative mineralization rate was the highest of all treatments. Thus, the change of soil structure plays an important role in organic carbon mineralization. The change in mechanical properties of the soil caused by the change of soil structure remains to be further studied.

In addition to soil structure, texture, and nutrients, soil organic carbon mineralization also has an impact on water content. In our study, the organic carbon retention effect in the C2 treatment was the best. The high moisture content may have little effect on the amplitude variation, similar to the variation law revealed by Jia et al. (2017), who believed that the cumulative mineralization amount and mineralization rate of organic carbon increase with the increase of soil moisture content but then subsequently gradually decreases. In our previously reported experiments (Sun & Han, 2018), we showed that the C2-like treated compound soil had the highest water content and the C1-like treatment had the greatest impact on soil water storage. According to Cooper et al. (2011), temperature was the primary factor driving soil organic carbon mineralization. The mineralization rate and cumulative mineralization amount of soil organic carbon increased with the increase of cultivation temperature, though the most significant effect of the temperature observed was outside the normal temperature range. Among the different agricultural measures, tillage systems also change the stability of soil aggregates, thereby affecting the loss of carbon stocks. Stable macroaggregates in cultivated soil can retain more carbon than microaggregates but macroaggregates were more easily mineralized than microaggregates (Goh, 2004). The soil also contains many different metal ions. One study indicated a significant negative relationships between Fe-oxyhydroxides, Al-oxyhydroxides and Al-humus complex content, and soil C mineralization, suggesting a mineral control of C mineralization (Rasmussen, Southard & Horwath, 2006). Therefore, the adsorption of these soil minerals can prevent the microorganic decomposition of organic matter. Taken together, soil nutrients, texture, water content, temperature, tillage measures, metal ions, and many other factors cause differences in the soil organic carbon mineralization. Future research should therefore focus on the later stages of the factors not involved.

Soil potentially mineralizable organic carbon (C0), also known as biodegradable carbon, is the total amount of organic matter that can be decomposed under the action of microorganisms (Guo et al., 2019). The C0 values in this study were consistent with the changes in Ct value, and the specific performance was C3>C1>C2>CK. The reason was that with the increase in soft rock content, the non-capillary space between the sand grains was filled by the soft rock, increasing the capillary pressure and promoting the formation of the soil aggregate structure. The soft rock was also rich in carbonate and other mineral components. As the soft rock volume fraction increased, the cementation force of the compound soil also increased significantly. Because the organic carbon content of the compound soil is significantly higher than that in the CK treatment, the activity of plant roots and animals in the compound soil also promotes the fusion of soft rock and sand (Han, Liu & Luo, 2012). Li, Qiu & Zhang (2010) studied the soil organic carbon mineralization in the Loess Plateau. They found that that the organic carbon mineralization rate constant k was neither affected by soil nutrient nor by pH, but it was influenced by particle composition. The results in our study showed that there was no significant difference in the k-value between the C1, C2, and C3 treatments, though they were significantly lower than in the CK treatment. Also, the changes in T1∕2 and k values were opposite. One explanation might be that the long-term application of chemical fertilizers in this experiment increased the inorganic nitrogen content, such as soil nitrate nitrogen and ammonium nitrogen, which then reacts with other compounds such as lignin or phenol present in the soil. This reaction lowers the decomposition properties changed of organic carbon had (Jenkinson, Fox & Rayner, 1985; Liu, Du & Li, 2017). Other studies showed that the increased amount of soft rock can promote the formation of aggregates in the compound soil, so that some organic carbon particles are encapsulated by the aggregates thus avoiding degradation and increasing the retention time of organic carbon in the soil (Pulleman & Marinissen, 2004; Chevallier et al., 2004). The C0/SOC value can reflect the solid storage capacity of the compound soil organic carbon: the larger the ratio, the stronger the soil organic carbon mineralization ability, and vice versa. The results of this study indicated that the C0/SOC values in all treatments were C3>CK>C2>C1, with no significant difference between C1 and C2, which was consistent with the trend observed in the soil organic carbon accumulation mineralization rate.

Conclusions

The soil organic carbon content can be significantly increased by the different compound ratios of soft rock and sand. With an increased soft rock content, the content of soil sand gradually decreased, while the content of clay and silt gradually increased, with the largest increase in silt. The soil texture changed from sand to sandy loam, then to loam and silty loam. The results of the scanning electron microscopy showed that the specific surface area between large particles and small particles increased with the increase of volume fraction between soft rock and sand. Interestingly, when the soft rock volume fraction was 50%, the soil structure collapsed. The C1 and C2 treatments had the highest mineralization rate on the first day of incubation, whereas the CK and C3 treatment reached their maximum on the third day of cultivation. The whole cultivation process can be divided into a rapid decline between days 1 to 11 and a slow decline between days 11 to 30. With the prolongation of cultivation time, the accumulation intensity of cumulative mineralization of soil organic carbon was gradually reduced. The cumulative mineralization rate in the C1 and C2 treatments was the lowest in all treatments, and C0/SOC was consistent with its variation rule. The organic carbon turnover rate was significantly decreased and the retention time in soil was increased with the addition of soft rock. Here, the C1 and C2 treatment showed the best effect. The accumulation of compound soil organic carbon can be significantly increased when the ratio of soft rock to sand was 1:5 to 1:1. A comprehensive mineralization index and scanning electron microscopy analysis indicated that the compounding ratio of 1:5 to 1:2 can be used as an important basis for farmland carbon sequestration and soil remediation measures.

Supplemental Information

Data S1 Regression equation of soil organic carbon mineralization rate and cumulative mineralization under different compounding ratios

CK: the volume ratio of soft rock to sand is 0:1; C1: the volume ratio of soft rock to sand is 1:5; C2: the volume ratio of soft rock to sand is 1:2; C3: the volume ratio of soft rock to sand is 1:1.

Click here for additional data file.

Data S2 Cumulative mineralization of SOC after the 30 days of incubation and parameters of its kinetic equations

CK: the volume ratio of soft rock to sand is 0:1; C1: the volume ratio of soft rock to sand is 1:5; C2: the volume ratio of soft rock to sand is 1:2; C3: the volume ratio of soft rock to sand is 1:1. Ct for amount of organic carbon cumulative mineralization, C0 for amount of potential mineralizable organic carbon, k for constant of organic carbon mineralization rate, T1∕2 for half turnover period, C0/SOC for ratio of potential mineralizable organic carbon to total organic carbon in compound soil. Values followed by different letters in the same column mean significant difference at 0.05 level between treatments, ** indicates a extremely significant level of 1%.

Click here for additional data file.

Data S3 Fig 1 Raw data: Test plot layout for soft rock and sand compound soils

Test plots 1 to 15 were set in 2009 and 16 to 30 were set in 2016. Plots 2, 7, 15, 17, 22 and 30 show the volume ratio of loess to sand is 1:2, plots 6, 10, 13, 21, 25 and 28 show the volume ratio of soft rock to sand is 1: 1, plots 5, 8, 9, 20, 23 and 24 show the volume ratio of soft rock to sand is 1: 2, plots 1, 4, 12, 16, 19 and 27 show the volume ratio of soft rock to sand is 1: 5, plots 3, 11, 14, 18, 26 and 29 show the volume ratio of soft rock to sand is 0:1. The color-marked test plots are selected for this trial. The red area represents CK treatment, the purple area represents C1 treatment, the blue area represents C2 treatment, and the yellow area represents C3 treatment.

Click here for additional data file.

Data S4 Organic carbon content of compound soils in different proportions of soft rock and sand

Different letters above the bars mean significant difference (at 0.05 level) between treatments. CK: the volume ratio of soft rock to sand is 0:1; C1: the volume ratio of soft rock to sand is 1:5; C2: the volume ratio of soft rock to sand is 1:2; C3: the volume ratio of soft rock to sand is 1:1.

Click here for additional data file.

Data S5 Organic carbon mineralization rate of compound soils in different proportions of soft rock and sand

CK: the volume ratio of soft rock to sand is 0:1; C1: the volume ratio of soft rock to sand is 1:5; C2: the volume ratio of soft rock to sand is 1:2; C3: the volume ratio of soft rock to sand is 1:1.

Click here for additional data file.

Data S6 Fig 3 Raw data: Organic carbon cumulative mineralization of compound soils in different proportions of soft rock and sand

CK: the volume ratio of soft rock to sand is 0:1; C1: the volume ratio of soft rock to sand is 1:5; C2: the volume ratio of soft rock to sand is 1:2; C3: the volume ratio of soft rock to sand is 1:1.

Click here for additional data file.

Data S7 Fig 4 Raw data: Cumulative mineralization rate of soil organic carbon under different compound ratios during the 30 days of incubation

CK: the volume ratio of soft rock to sand is 0:1; C1: the volume ratio of soft rock to sand is 1:5; C2: the volume ratio of soft rock to sand is 1:2; C3: the volume ratio of soft rock to sand is 1:1. Different letters above the bars mean significant difference (at 0.05 level) between treatments.

Click here for additional data file.

Data S8 Fig 5 Raw data: The microstructure of compound soil in different proportions of soft rock and sand

(a): the volume ratio of soft rock to sand is 0:1 (CK); (b): the volume ratio of soft rock to sand is 1:5 (C1); (c): the volume ratio of soft rock to sand is 1:2 (C2); (d): the volume ratio of soft rock to sand is 1:1 (C3). The magnification is 1000 times.

Click here for additional data file.

Data S9 Fig 6 Raw data: The soil particle composition under different compound ratios of soft rock and sand

CK: the volume ratio of soft rock to sand is 0:1; C1: the volume ratio of soft rock to sand is 1:5; C2: the volume ratio of soft rock to sand is 1:2; C3: the volume ratio of soft rock to sand is 1:1. Lowercase letters indicate significant differences (at 0.05 level) in the same particle composition between treatments.

Click here for additional data file.

We thank professor Jichang Han for providing the soil resources.

Additional Information and Declarations

Competing Interests

Author Contributions

Data Availability

All the authors are employed by Shaanxi Provincial Land Engineering Construction Group Co., Ltd., and all the authors declare that they have no competing interests.

Zhen Guo and Jichang Han performed the experiments, analyzed the data, prepared figures and/or tables, authored or reviewed drafts of the paper, approved the final draft.

Yan Xu and Yangjie Lu performed the experiments, contributed reagents/materials/analysis tools, approved the final draft.

Chendi Shi and Lei Ge approved the final draft.

Tingting Cao and Juan Li conceived and designed the experiments, approved the final draft.

The following information was supplied regarding data availability:

The raw measurements are available in the Supplemental Files.

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
