# Peer review of "The mineralization characteristics of organic carbon and particle composition analysis in reconstructed soil with different proportions of soft rock and sand"

_PeerJ, doi:10.7717/peerj.7707_

## Round 0.1 · original submission · Major Revisions

As you see below, both reviewers found that your manuscript addresses an important and interesting topic in terms of how organic carbon mineralization rate differs between the soft rock and sand mixture. However, I agree with one of the reviewer’s claim that the English language and the citation needs to be very much improved. And most importantly, you want to add more detailed discussion about the factors determining the observed differences in mineralization rates. Finally, as both reviewers pointed out, I also request additional explanation for the time scale of your experiments.

Reviewer 1 ·

Basic reporting

The article from Guo et al. is an interesting piece of work which shows organic carbon mineralization rate difference according to the soft rock and sand mixture. From organic matter decomposition experiment, the results show that carbon accumulation effectively increases when the ratio of soft rock and sand volume was about 1/5-1/2. Overall, I think this is a very nice manuscript with several important findings about carbon cycling in the soil.

Experimental design

no comment

Validity of the findings

no comment

Additional comments

*Does the amount of cumulative mineralization of soil organic carbon under different mixture differ only in mineralization rate? Or does the amount differ for long period experiment? Add to the discussion if there are something hypothesis in other papers discussion.

*According to Kemmitt et al. (2008), incubation for 30 days is only explained non-biologically available SOM. Discuss or Explain about this thing in discussion section or method section.


Specific comment
*Line 27: Replace the words “typical treatments” with such as “mixture of soft rock and sand”.
Line 77 and 149: What hm2 means?
Line 119 and 120: Which is correct for Fping or Fuping?
Line 220 C3?
Line 241: How to estimate the k?
L249: Define SOC.

Figures 1, 3, 4, and 6: After the decimal point is not necessary.
Figures 2 and 3: There are too many horizontal scale marks.

Reviewer 2 ·

Basic reporting

The English and the referencing needs to be improved (see general comments below)

Experimental design

The manuscript compares mineralization in 4 artificial soil plots, in triplicates. It does not offer an in-depth assessment of the factors responsible for the observed differences in mineralization rates (see general comments below)

Validity of the findings

no comment

Additional comments

The manuscript by Guo and colleagues reports on the mineralization of organic matter in soils composed of mixtures of 'soft rock' and sand. This is motivated by the importance of soil respiration under a changing climate. The soils were established in triplicate plots 2 m * 2m * 0.7 m plots (with the top 30 cm containing the mixture, overlying 40 cm of sandy soil) in 2009, and mineralization rates and soil characteristics were quantified in spring 2018.

The addition of soft rock (containing SiO2, Al2O3, Fe2O3 and CaO) increased soil carbon content, so that treatments with substantial soft rock addition (C3 (softrock/sand = 1:1), C2 (1:2), C1 (1:5)) are more enriched in organic carbon than those without (CK, softrock/sand = 0:1), albeit not in a linear fashion (Fig. 1). Treatments C1, C2 and C3 also exhibited higher rates of organic matter mineralization as measured over a 30 day period. However, the first order rate constants were largest for CK, where no soft rock was added.

Assessment: The manuscript addresses an important topic. However, it does not meet the bar required for publication. The introduction alludes to a theoretical basis, but the results presented are not meaningfully put into such a context. Curves are fitted through the data, but little rationale is given. The data mainly contains observations on organic carbon mineralization on 4 treatments over a time span of a month. It is not clear why the observational period over a month is the relevant time scale, and if/to what extent these results are relevant for longer timescales. The study lacks the data that allows one to link the observations to the underlying processes (e.g. measurements of sorption). It refers to microorganisms (line 284) or different organic matter compounds (line 295), but there is no data to back up such statements. Similarly, the discussion of extracellular processes and hydrolysis of organic matter seams reasonable, but there is no supporting evidence provided (line 303).

Clarity: The manuscript requires revision to clarify the language. For example, line 69 ff "... weathering and .. soil are erosion and texture...", line 76 "the technology ... has realized the resource utilization....", or line 87 "the scholars have carried on resource utilization and obtained the certain achievement" all need to be reworded.


Referencing: some broad statements lack references (e.g., end of line 62, or when referring to previous work (line 96/97,). Also, a significant fraction of the sources cited are journals in Chinese with English abstracts. This may be acceptable for very select sources that reflect local knowledge. However, it is also the case when citing methods, which prevents a significant portion of the readership from assessing the methodology (lye absorption method).

- The abstract contains references to treatments CK, C1, C2 and C3 that are not explained
- Throughout the manuscript, results are given with an accuracy that surely go beyond the significant digits (e.g. line 123 ff)
- Line 56 should be 'slightly changed'

---

## Round 0.2 · Major Revisions

The manuscript has been significantly improved but the reviewer is still raising a number of concerns. Based on the PeerJ's acceptance criteria, in particular, "Decisions are not made based on any subjective determination of impact, degree of advance, novelty, being of interest to only a niche audience, etc" and "Speculation is welcomed", I will welcome your resubmission if you could address all the points the reviewer raised except for "the scope is very limited" and "discussion is not or only loosely tied to data".

Reviewer 2 ·

Basic reporting

see comments below. The writing has been improved in the revision

Experimental design

see comments below. I have some concerns on the lack of data that lead to a system understanding, and the meaningfulness of potential mineralization rates measured

Validity of the findings

see comments below

Additional comments

The revised manuscript by Guo et al. has been improved by revisiting the writing, and by expanding the references and the discussion

The editing indeed made the manuscript easier to read. However, it still suffers from numerous issues. Most importantly, the scope is very limited: it reports on the amount of organic matter that can be mineralized in soils made from different mixtures of soft rock and sand. Ancillary measurements (granulometry, SEM) provide some limited context, but what is lacking is a data-supported analysis of the processes underlying the limited set of observations. The discussion is not or only loosely tied to data in this manuscript and hence is largely speculation. There seems to be some additional data that support some of the comments made, but I was unable to access or read them (e.g. paper by Jia et al, Zhang et al, Li et al). Thus, overall, the paper offers some novel data, but it does not provide much insight into the processes that cause the observations.

introduction
- when referring to 'culture', do you mean 'experiment' or 'incubation'?
- What does it mean that "the study of organic carbon mineralization... can provide technical support ... for soil organic reconstruction"?
- "tIn summary, when the ratio of soft rock to sand volume was 1:5-1:2, this can effectively increase the accumulation of soil organic carbon. " In fact, figure 1 shows that there is more OC in C1 than in C2; figure 2 shows that the rate betwee 11-30 days - which might be an indicator of the longer term behavior is highest, not lowest in C3; C3 also has the highest cumulative mineralization (Fig 3, Fig4)
- "Then, the distribution of soil particles was more uniform, the soil structure was stable, and the mineralization level of unit organic carbon was lower." Where in the paper do you show soil stability? This is not the same as microstructure, so what data are you referring to?

Line 68: loose -> loss
Line 73: optimum proportion in what sense? what is your metric?
Line 83: "Many scholars have made use of the low-efficient resources like soft rock and sand and achieved some interesting results. " Delete or refer to actual studies.
Line 84-88: the role of soils in C storage/loss not only depends on the emission of CO2 but also the amount being fixed.
Line 90: adsorption potential
Line 94: "The texture was also improved to a certain extent" - improved in what sense?
"and the macroscopic mechanics showed a strain hardening phenomenon with non-linear characteristics" - what data is this based on?

methods:
- what is the motivation for having the sample "pre-cultured in the incubator at 25degC for 5 days"? How does that impact the soil and the subsequent results?
- "For uniformity of factors such as illumination and micro-topography, the test plot was arranged from south to north in a “one” shape". It is not quite clear how this was set up. Was there no random distribution of different plots?
- Mineralization rates quantified by an alkali absorption method over a short (30d) time horizon.
I am not quite following the rationale given, which refers to prior work by Dai et al. also "set the incubation time is 30 d), the results are satisfactory, and the test method can be repeated". How does this short timescale connect with the climate change issue referred to in the introduction? Also, how does the measurements of potential C mineralization carried out in this manuscript relates to actual fluxes (or rates) in the field?

data analysis:
cumulative mineralization (M, presumably in mg/kg, see Fig3, though y in table 1 is given in mg/kg/d) and the instantaneous mineralization rate (R, mgCO2/kg/d) are described by a logarithmic function: M, R = a + b*ln(t). However, Table 2 and eq. 1 suggest a first order dependency of R on the organic matter concentration. How are these different fits reconciled in a conceptual kinetic model of organic matter breakdown?

discussion:
as mentioned above, there is no or little data presented that supports the model (non-biological vs. biological controls). It is also not clear to me how well the soils of this study compare to eg. those investigated by Kemmitt et al.
Line 370 ff also does not really add much new knowledge.
Line 417/8: the accumulation of SOC is larger fro C3 (1:1) than C2 (1:2) - see figure 1

references: fix title for She et al.

---

## Round 0.3 · Minor Revisions

Although the article is acceptable, I encourage you to include the photo with the table showing the experimental setting in your rebuttal letter is included in the main text as a new figure. I also recommend that you try to reflect the replies against the reviewer's comments to the main text as succinctly as possible. Otherwise, this article can be accepted - so please consider these suggestions and resubmit

---

## Round 0.4 · accepted · Accept

Thank you very much for your further revision.